# Trends in malaria prevalence among school-age children in Mainland Tanzania, 2015–2023: A multilevel survey analysis

**Frank Chacky**[1,2,3]*, **Joseph T. Hicks**[4], **Mbaraka John Remiji**[5], **Susan F. Rumisha**[5,6,7], **Patrick G.T. Walker**[4], **Prosper Chaki**[5], **Sijenunu Aaron**[1,2], **Samwel L. Nhiga**[1,2], **Erik Reaves**[8], **Naomi Serbantez**[9], **Fabrizio Molteni**[10,11], **Billy Ngasala**[12], **Achyut KC**[12,13,14], **Bruno P. Mmbando**[15], **Robert W. Snow**[16,17], **Jean-Pierre Van Geertruyden**[3]

**1** Ministry of Health, Dodoma, Tanzania, **2** National Malaria Control Programme, Dodoma, Tanzania, **3** Global Health Institute, University of Antwerp, Antwerp, Belgium, **4** MRC Centre for Global Infectious Disease Analysis, School of Public Health, Imperial College London, London, United Kingdom, **5** Department of Environmental Health and Ecological Sciences, Ifakara Health Institute, Dar es Salaam, Tanzania, **6** Department of Public Health, Biostatistics and ICT, National Institute for Medical Research, Dar es Salaam, Tanzania, **7** Telethon Kids Institute, Malaria Atlas Project, Nedlands, Western Australia, Australia, **8** United States of America President's Malaria Initiative, Centers for Disease Control and Prevention, Dar es Salaam, Tanzania, **9** United States of America President's Malaria Initiative, United States of America Agency for International Development, Dar es Salaam, Tanzania, **10** Swiss Tropical and Public Health Institute, Dar-es-salaam, Tanzania, **11** Swiss Tropical and Public Health Institute, Basel, Switzerland, **12** Department of Parasitology and Medical Entomology, Muhimbili University of Health and Allied Sciences, Dar es Salaam, Tanzania, **13** Department of Microbiology, Tumor and Cell Biology, Karolinska Institutet, Stockholm, Sweden, **14** Ubuntu Health, Atlanta, Georgia, United States of America, **15** Department of Public Health, Biostatistics and ICT, National Institute for Medical Research, Tanga Center, Tanga, Tanzania, **16** KEMRI-Wellcome Trust Research Programme, Nairobi, Kenya, **17** Nuffield Department of Medicine, Centre for Tropical Medicine & Global Health, University of Oxford, Oxford, United Kingdom

* frank.chacky@gmail.com, frank.chacky@afya.go.tz

## Abstract

In high-transmission areas, school-aged children have higher malaria prevalence and contribute significantly to the transmission reservoir. Malaria infections can be asymptomatic or present with symptoms which may contribute to anaemia, severe illness and fatal malaria. This analysis provides trends of malaria prevalence and associated risk factors among school-aged children in mainland Tanzania. Data for this analysis were obtained from nationwide school malaria surveillance conducted every other year from 2015 to 2023. A total of 307,999 school children aged 5–16 years old from 850 public primary schools were tested for malaria infection using rapid diagnostic tests, assessed for malaria control intervention coverage and other malaria-related parameters. A multilevel mixed-effects logistic regression model was used to assess associated risk factors. Overall malaria prevalence was 21.6% (95%CI: 21.3–22.0) in 2015 which progressively decreased to 11.8% (95%CI: 11.5–12.0 p <0.001) in 2021 with no significant change in the overall malaria risk between 2021 and 2023 (AOR 1.32, CI: 0.92–1.81, p=0.08). School children aged between 9–12 years and 13–16 years had 20% higher risk of malaria (95% CI: 1.15–1.25) and 21% higher risk of malaria (95% CI: 1.16–1.27), respectively, compared to those aged between 5–8 years. Geographically, children from the Lake zone had the highest odds of prevalence (AOR: 18.75; 95% CI: 12.91–27.23) compared to the Central zone, and

**Data availability statement:** The data used to generate these outputs was obtained from the national school-based surveillance system established in 2014-15. This is primarily programmatic surveillance data, rather than research data. Most of the data used in this analysis is presented in the respective figures, tables, and supplementary information. Detailed reports from the previous surveys of the Tanzania SMPS can be accessed at nmcp. go.tz/index.php/publication. However, given the current situation and limitation to sharing government data, any requests for additional data supporting the findings for this study can be made to the Tanzania Ministry of Health at https://www.moh.go.tz/contact or send email directly to the Permanent Secretary at ps@ afya.go.tz.

**Funding:** Phase I of Round 1 of the School Malaria Parasitemia Survey, which covered five regions, was funded by the UK Department for International Development (DfID) through the University of Oxford as part of the "Strengthening the Use of Data for Malaria Decision Making in Africa" program (DfID Program Code #203155). Funding for the remaining regions in Round 1 and subsequent surveys was provided by the Global Fund to Fight AIDS, Tuberculosis, and Malaria (GFATM) under the following grant numbers: TZA-M-MOFP-P03 (4034) for the 2015 and 2017 survey rounds, TZA-M-MOFP-P04 (4276) for the 2019 round, and TZA-M-MOFP-P05 (4688) for funds received to support the 2021 and 2023 survey rounds (https://www. theglobalfund.org/en/). The U.S. President's Malaria Initiative (PMI), through the U.S. Agency for International Development's Okoa Maisha Dhibiti Malaria program (Cooperative Agreement no. 72062118CA-00002), implemented by RTI International in partnership with the U.S. Centers for Disease Control and Prevention, provided financial support for data management and report writing (https:// www.pmi.gov/where-we-work/tanzania/). RWS was supported as a Wellcome Trust Principal Fellow (#212176) and acknowledges additional support from the Wellcome Trust for the Kenya Major Overseas Programme (#203077). The funders played no role in the study's design, data collection and analysis, decision to publish, or manuscript preparation. None of the authors were supported by funds used to finance this study from either the Global Fund or PMI/CDC via RT.

**Competing interests:** The authors have declared that no competing interest exist.

sleeping under an insecticide-treated net demonstrated a protective effect (AOR=0.68, 95%CI: 0.64–0.72, p < 0.001). There was a significant decline in the prevalence of malaria infection across the study period. We presented a countrywide active surveillance data, collected over time and in different settings which are unique and seldom presented. We believe various stakeholders will use our findings and join force to combat malaria not just in Tanzania but, in all malaria endemic countries.

## Introduction

The burden of *Plasmodium falciparum* malaria infection has almost halved following the scale-up of malaria control interventions since the year 2000, particularly in children under-five years of age in sub-Saharan Africa [1]. Following the large-scale implementation of malaria control interventions to biologically vulnerable groups (pregnant women and under-fives), there has been a marked difference in the burden of malaria among younger and older school-age children [2–4]. Malaria infection in school children is associated with increased risks of anemia, reduced ability to concentrate and learning, impaired cognitive function, school absenteeism, and reduced academic achievement [5–7]. The burden of malaria infection in this age is a threat to their health and education as well as for onward transmission of *P. falciparum*. However, the progress in protecting older school-age children from symptomatic and asymptomatic falciparum malaria remains challenging [8–10]. Currently, no specific policies are in place to address the high burden of the prevalence of malaria infection in school-age children, as universal coverage of insecticide-treated bed nets (ITNs) and access to prompt diagnosis and treatment have been assumed to cover this age group [9].

In Tanzania, school-age children have high malaria infection prevalence [11,12]. In 2014, the Tanzania School Net Programme (SNP) was introduced as a keep up strategy to contribute attaining universal coverage of insecticide treated nets (ITNs), with annual phased distribution to children attending schools in the selected primary grades [13,14]. To improve the understanding of malaria prevalence at the sub-national level among school-age children, which has not been possible through other population-based surveys such as the Demographic Health Surveys and Malaria Indicator Surveys [15], Tanzania strengthened its surveillance in school children through the School Malaria Parasitological Survey (SMPS). School-based surveillance of malaria provides strong epidemiological basis for planning as well as monitoring and evaluating malaria control interventions [16] across different malaria transmission settings [16–19]. The SMPS obtains data on malaria infection, uptake of malaria control interventions, and other malaria-related parameters to guide informed strategies. In this study, we aimed to determine malaria infection prevalence and associated risk factors among school children in mainland Tanzania from 2015 to 2023.

## Materials and methods

### Study setting

Tanzania with an area of 945,000 km$^2$, is administratively divided into regions, councils, wards, and villages (Fig 1). Two to seven regions form a zone, totaling eight geographic zones in mainland Tanzania which are used to estimate geographic differentials of demographic and health indicators by the Ministry of Health [20]. According to the 2022 National Population and Housing Census (NPHC), the population of mainland Tanzania was 59,851,347 with about 31.6% being children between 5 and 16 years of age group [21].

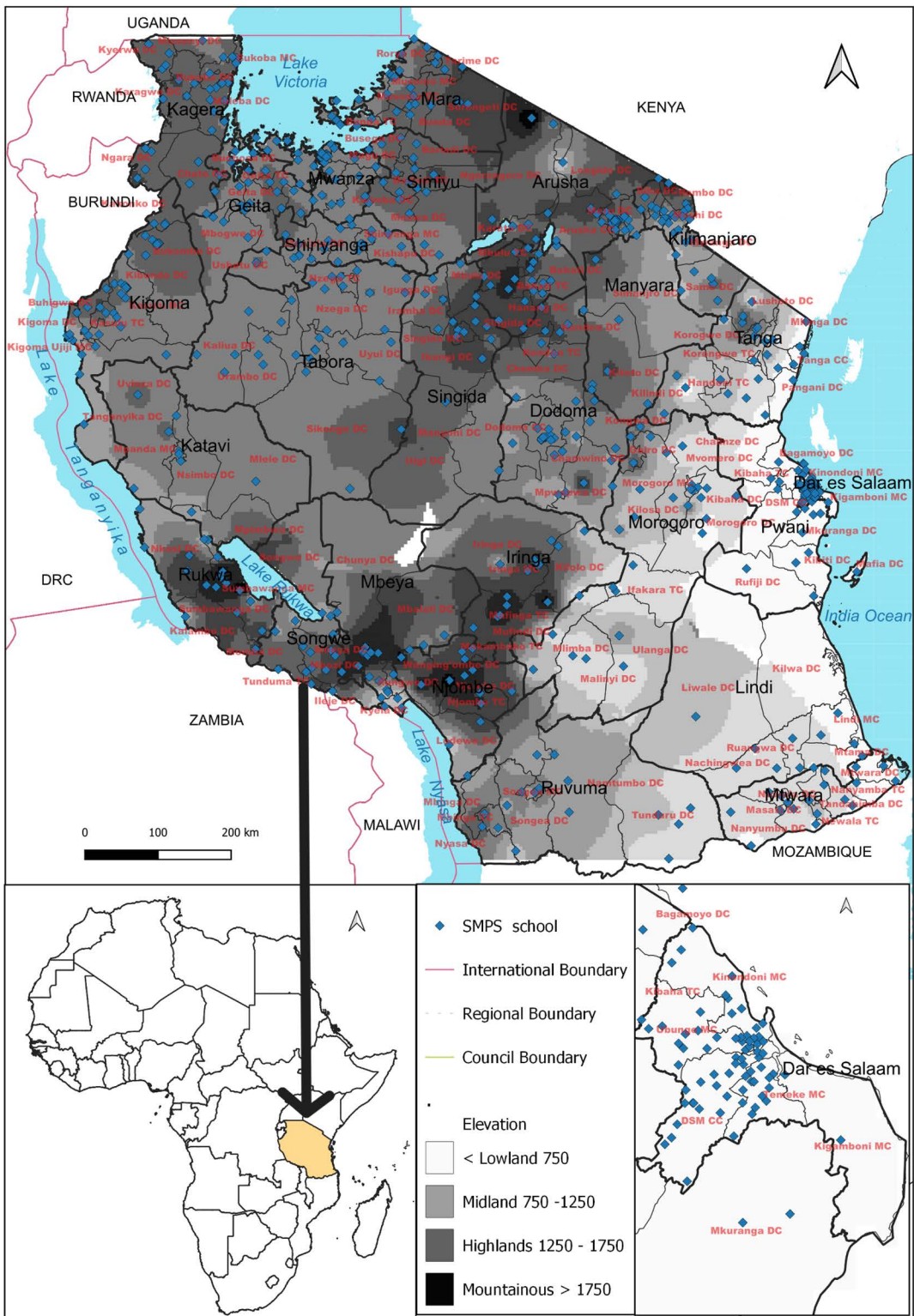

**Fig 1. A map of Tanzania indicating surveyed schools along with regional and council boundaries was created.** The base map was generated using a shapefile for Tanzania that included regional and council boundaries, obtained from NBS Microdata Catalog. Geographic coordinates (latitude and longitude) of the surveyed schools were recorded during data collection. QGIS software was used to plot and visualize the data on the map.

Tanzania is characterized by a tropical climate with regional variations shaped by its diverse and complex topography. The coastal lowlands extend approximately 150 kilometers inland from the Indian Ocean, reaching altitudes of about 300 meters (m) above sea level (asl). These regions are warm and humid, the temperature rarely falls below 20°C throughout most of the year. Lake Victoria basin has higher temperatures, humidity, and heavier rainfall. In the Central Plateau mountain ranges are about 900 to 1,800 m asl, with diurnal temperature fluctuations, warm to hot days and cool nights, a short rainy season, and a prolonged dry period. The Northern and Southern Highlands, have temperate climates, with temperatures averaging around 20–23°C year-round, except during the cool season (June–September) when temperatures may drop below 17°C. These diverse ecological settings significantly influence malaria transmission.

About 93% of the Tanzanian population is at risk of malaria; however, the transmission intensity is heterogeneous across the different geographic zones. In the arid central plateau malaria transmission is unstable and seasonal; in the southern part it is stable and seasonal; and the northern and western parts it is bimodal, and stable throughout the year in the coastal fringe, southern lowlands, and the Lake Zone.

## Survey design

The SMPS is an ongoing biennial cross-sectional school survey of children aged 5–16 years enrolled in the public primary schools from all the 184 councils of mainland Tanzania, which was first conducted in 2014–2015 [15].

A multistage, stratified, proportional probability to size sampling approach was used. To address malaria transmission heterogeneity, each council was stratified into several strata based on altitude (low and highland areas), population density, urbanity (urban vs rural), malaria endemicity and other topographical features. In each council, the required number of school children was calculated based on the respective population size and previous malaria prevalence, except in the first SMPS where the population-weighted malaria prevalence rate adjusted for children aged 2–10 years ($PfPR_{2-10}$) for 2010 was used [22] (S1 Fig).

In the first stage, one ward was randomly selected from each stratum as the primary sampling unit (S1A Fig).

In the second stage, one school was randomly selected as the secondary sampling unit from the ward based on probability proportion to size. School selection probability depended on the number of wards and schools within the strata in the council (S1A Fig).

In the third stage, school children were selected from the school by using systematic sampling from the class registers with a 1:1 ratio for females and males. Each class (standards 1 to 7) was allocated study participants based on the proportion of the school size (S1B&S1C Fig). The minimum and maximum number of children sampled per school was 60 and 120 respectively.

The sampling strategy was designed to establish a school-based surveillance system in the country, where the lowest unit of analysis is the school rather than individual children. Where possible, school selection was restricted to one school in villages with more than one school. In the 2019, 2021 and 2023 survey rounds, the SMPS maintained the same schools surveyed previously unless individual councils raised concerns due to changes in geographical settings, such as the formation or shifting of wards from one council to the other.

Using Microsoft Excel, the sample size was calculated assuming a margin of error of 0.05, a 95% confidence interval, and a design effect of 2.5 to account for cluster sampling and regional heterogeneity. The number of schools for the survey in each council was determined based on council sample size, assuming an average of 100 children per school.

## Survey team and Affiliations

Data collection was carried out by a team from respective councils, including the malaria focal person, two laboratory technicians, one education officer, and two teachers from the surveyed schools. In each region, a regional team included malaria focal person and laboratory technician and a national supervisor from either the National Malaria Control Programme (NMCP), National Institute for Medical Research (NIMR), President's Office Regional Administration and Local Government (PORALG), National Bureau of Statistics (NBS), Sokoine University of Agriculture (SUA), University of Dar es Salaam (UDSM), Muhimbili University of Health and Allied Sciences (MUHAS), Hubert Kairuki Memorial University (HKMU) or Ifakara Health Institute (IHI).

## Survey team orientation and fieldwork exercise

Three-day orientation workshops for field teams and national supervisors were organized in four designated training centers. The training included orientation of the team to study protocol, malaria testing protocol, mRDT standard operating procedures manual, job aide for fieldwork procedures, sampled children consenting procedure, survey tools, quality assurance and quality check procedures, handling field forms, and the use of electronic data collection tools including handling and data management.

The council team conducted community sensitization (children, parents, teachers, village leaders, and school committees), sampling, interviewing, malaria testing and interpretation of the test results, provision of antimalarial drugs to children who tested positive for malaria parasites, performed referral to the nearby health facilities to any child found sick at the school, and handling of the materials and equipment used. Respective school teachers assisted in recording children's identification numbers and distributing refreshments. Regional teams and national supervisors were responsible for supervising the respective regions on quality checks of used malaria rapid diagnostic tests (mRDTs), filled forms, perform redistribution of supplies where needed, and the collection of filled tools and financial documents to the central level.

## Data collection period

The SMPS which is a cross-sectional, school-based malaria survey was conducted in five rounds from 2015–2023.

Round I survey was carried out between the year 2014 and 2015 and conducted in three phases: phase I included 113 schools in 5 regions between August 11th and September 18th 2014; phase II was conducted in 11 regions from May 18th to June 1st in 2015 and included 217 schools; and phase III which included 207 schools was conducted between 7th and 16th October in 2015 in 9 remaining regions totaling 25 regions existed in mainland Tanzania in that time.

The 2017 SMPS included 615 schools and data collection was done between July 24th to November 13th. The 2019 SMPS data collected was conducted between 19th of August to 1st September and included 646 schools. The 2021 SMPS included 687 schools and data collection started on 2nd October to 24th November and the 2023 SMPS data collection which included 651 schools started in August 2nd to 14th October.: S1 Table provide details on the SMPS data collection periods.

## Data management

Data were collected electronically or paper-based (except in 2015, which was entirely paper-based). Socio-demographic and malaria-related information collected included the child's age, sex, history of fever, ownership, and use of bed nets a night preceding the survey. Individual

questionnaires were captured electronically while designated mRDT registers for malaria testing were collected in hard copy. Electronic data were sent directly to the national server hosted centrally by the Ministry of Health. Field supervisors and investigators reviewed data daily at the end of each day. Paper-based data were single-entered by a group of trained and experienced data clerks, using EpiData (EpiData Association, Denmark) templates and using mobile applications data were entered directly into Composite Management Information System (CMIS) in the 2023 survey round under the guidance of study investigators, developers, and statisticians.

Data merging including individual records, schools, councils, and regions with respect to different survey rounds was conducted to address naming and duplication issues.

## Variable definitions

### Dependent variable

The dependent variable was malaria infection, defined as a positive rapid diagnostic test (mRDT). In 2014/2015, a specific mRDT detecting *Plasmodium falciparum* and *Plasmodium vivax* antigens was used, while a different mRDT detecting *Plasmodium falciparum* HRP2 and pan-plasmodial pLDH antigens was used in all subsequent surveys. Aggregated results were expressed as malaria infection prevalence, defined as the percentage of mRDT-positive schoolchildren.

### Independent variables

The factors used to explain the dependent variable are defined at two-levels, which are

### Individual-level variables

School children were categorized into three age groups: 5–8, 9–12, and 13–16 years. Sex was categorized into male and female. Children were asked about history of fever during the 14 days before the survey, sleeping under a bed-net the night before the survey and mosquito bed-net ownership in their households.

### School-level variables

School elevation refers to the altitude of the area relative to sea level measured in meters and grouped into four categories: 0–<750 (lowland), 751–<1,250 (midland), 1250–1,750 (highland), and ≥ 1751 (mountainous). To estimate geographic differentials in demographic and health indicators; mainland Tanzania regions were grouped into eight geographic zones as follows: Central (Dodoma, Singida and Manyara regions), Eastern (Dar es Salaam, Pwani and Morogoro), Lake (Kagera, Mwanza, Geita, Mara and Simiyu, Shinyanga), Northern (Kilimanjaro, Tanga and Arusha), Southern (Lindi and Mtwara), Southern Highlands (Iringa, Njombe and Ruvuma), Southwest Highlands (Mbeya, Rukwa, Katavi and Songwe), and Western zone (Tabora and Kigoma).

### Data sources and measurements

The data for our study was extracted from the School Malaria Survey database. The information collected encompasses various demographic and environmental variables. Age (categorized as 5–8, 9–12, 13–16, or marked as Missing based on age range and observed patterns) and sex (classified as Female, Male, or Missing), were captured using a structured questionnaire tool.

Health-related variables, such as the presence of fever within the past two weeks (categorized as Yes, No and Missing) and sleeping habits regarding bed-net usage on the night before the survey (categorized as Yes, No and Missing), were also recorded. The missing data for history of

fever and sleeping under a bed net the night before the survey were attributed to the survey design applied in 2019, 2021, and 2023, where only a subsample of SAC was interviewed for additional information. Consequently, history of fever and sleeping under a bed net the night before the survey were not included in the modeling. Additionally, the dataset includes information on school elevation, with altitude ranges from 0 to <750m above sea level, 750 to <1250m, 1251 to 1750m, and>1750m. The geographical zones to which the schools belong were categorized as Central, Eastern, Lake, Northern, Southern, Southern Highlands, and Southwest Highlands.

## Statistical methods

The unit of analysis for this study was schoolchildren aged 5–16 years old. We pooled data from the SMPS database and exported to R software version 4.4.2. Data groupings were performed for continuous and categorical variables respectively. Numerical variables were summarized using medians with interquartile range (IQR), and categorical variables were presented using frequencies and percentages with 95% confidence intervals (95%CI).

In this study, variation within the dataset was assessed using statistical measures including the Intra-class Correlation Coefficient (ICC), Median Odds Ratio (MOR), and Proportional Change in Variance (PCV). The ICC, representing the proportion of total variance attributable to between-cluster variance, was calculated for both the null model (adjusted ICC: 0.648) and the full model (adjusted ICC: 0.497). Additionally, the MOR, quantifying the median odds of the outcome variable between different clusters, was determined for the null model (MOR: 17.4792). The PCV, indicating the percentage change in variance between models with and without predictors, was computed as 46.3%, reflecting a substantial reduction in variance with predictor inclusion. Model comparison and selection were based on criteria including the Akaike Information Criterion (AIC), with the best-fitting model chosen accordingly. Assessment of multicollinearity among predictors was conducted using Variance Inflation Factors (VIF), Standard Error (SE), and Variance Correlation Estimator (VCE). These analyses collectively provided insights into clustering, heterogeneity, and model fit within the dataset.

We conducted a multilevel mixed-effects logistic regression analysis to assess malaria infection risk factors, accounting for clustering by school with random intercepts for each school location. This model included both individual-level (age group, and sex) and school-level predictors (elevation, geographic zone, survey year) as independent variables. Analyses and visualizations were performed in R statistical software version 4.4.2. Statistically significant predictors (p-value ≤ 0.05) were identified, ensuring robust conclusions.

The statistical formula for the multilevel mixed-effect logistic regression model used in this study was:

$$logit(p) = \left(\frac{p}{1-p}\right)$$
$$= \beta_0 + \beta_1 \cdot Age_{group} + \beta_2 \cdot sex + \beta_3 \cdot Elevation_{group} + \beta_4 \cdot Zone$$
$$+ \beta_7 \cdot Survey_{year} + \mu_i$$

where:

- $P$ is the probability of having malaria (1 = positive, 0 = negative).

- logit(p) is the log-odds of the probability of having malaria.

- β0 is the intercept of the model.

- β1, β2, …, β7 are the coefficients for each predictor variable.

- Ui represents the random effect associated with the schools (where i indexes the schools).

### Ethical approval and informed consent

The five SMPS surveys were approved by the National Health Research Ethics Committee, which is a sub-committee of the Medical Research Coordinating Committee of the National Institute of Medical Research with reference numbers NIMR/HQ/R.8c/Vol.I/1715 (2014), NIMR/HQ/R.8a/Vol.I/373 for extension (2015), NIMR/HQ/R.8a/Vol.IX/2527 (2017), NIMR/HQ/R.8a/Vol.ix/3171 (2019), NIMR/HQ/R.8c/Vol.I/1857 (2021) and NIMR/HQ/R.8c/Vol.I/2354 (2023). The Principal Investigator and some Co-Investigators obtained certificates for the protection of human subjects and research information prior to the surveys. Other Co-Investigators, national supervisors, and field staff were trained using standard operating procedures on how to ensure the protection of human subjects.

The survey applied a passive, opt-out approach for parental permission approval whereby once children were selected for the survey during the preparatory days; a memo was written by respective school teachers to parents/guardians. It was assumed that parents/guardians approved their children's participation if they did not express their disapproval. In addition, approval was sought from each level before and during fieldwork exercise including permission letter from President's Office, Regional Administrative and Local Government (PO-RALG); written school committees' approval, and verbal assent from all surveyed children aged above 8 years old.

A child with a positive test result was treated with artemether–lumefantrine (ALu), as recommended in the National Malaria Diagnosis and Treatment Guidelines [1] and when deemed necessary was referred to the nearest health facility.

## Results

### Demographic characteristics of the survey participants

In the five cross-sectional surveys conducted in 2015, 2017, 2019, 2021 and 2023; a total of 307,999 school children from 850 public primary schools in mainland Tanzania were enrolled (Table 1). Overall, the median age was 10.6 (IQR–6 years old) years with the majority of participants in the 9 to 12-year-old age group, comprising 50.7% to 54.1% of the total population surveyed. The sex ratio remained similar across the surveys, with females accounting for approximately 50%. The majority of the surveyed children (ranging from 88.3% in 2015 to 88.8% in 2023) reported sleeping under an ITN the night before the survey. History of fever in the past two weeks before the survey, was highest (32.6%) in 2015 and lowest (11.2%) in 2023 (Table 1).

### Malaria infection prevalence

Malaria prevalence was 21.6% (95%CI:21.3%–22.0%) in 2015 and progressively declined to 11.3% (95% CI:11.0%–11.5%) in 2023. However, there was no statistically significant difference between the prevalence in 2021 (11.8%, 95% CI 11.5–12.0) and 2023 (11.3%, 95% CI:11.0–11.5) (Fig 2 and S2 Table).

Children aged 13–16 years had the highest malaria prevalence, ranging from 23.3% (95%CI: 22.6%–24.1%) in 2015 to 14.6% (95%CI:14.0%–15.2%) in 2023. Children aged 5–8 years had the lowest prevalence across all the survey years, ranging from 18.4% (95%CI 17.6%–19.2%) in 2015 to 8.6% (95% CI 8.2%–9.1%) in 2023 (Fig 2 and S2 Table).

In 2015, male children had a prevalence of 23.3% (95%CI 22.7%–23.8%) compared to 20.0% (95%CI:19.5%–20.5%) among female children which declined to 12.4% (95%CI:12.0%–12.7%) among male children and 10.1% (95%CI:9.8%–10.5%) among female children in 2023 (Fig 3 and S2 Table).

**Table 1. Socio-demographic characteristics of the survey participants 2015–2023.**

| Characteristic | Survey year | | | | |
|---|---|---|---|---|---|
| | 2015 | 2017 | 2019 | 2021 | 2023 |
| Number of Schools surveyed | 537 | 615 | 646 | 687 | 651 |
| Numbers of school children | 49,113 | 64,621 | 63,806 | 64,959 | 65,500 |
| Age years median (IQR) | 11.0 (9,13) | 11.0 (9,13) | 11.0 (9,13) | 10.3 (8.4,12.2) | 10.0 (8,12) |
| Age group (years), n (%) | | | | | |
| 5–8 | 9,242 (18.9) | 11,230 (17.8) | 14,643 (23.0) | 12,074 (19.2) | 19,110 (29.8) |
| 9–12 | 26,469 (54.1) | 33,926 (53.8) | 32,315 (50.7) | 34,102 (54.1) | 32,518 (50.8) |
| 13–16 | 13,232 (27.0) | 17,907 (28.4) | 16,781 (26.3) | 16,871 (26.8) | 12,405 (19.4) |
| Missing | 170 | 1,558 | 67 | 1,912 | 1,467 |
| Sex, n (%) | | | | | |
| Female | 24,728 (50.3) | 33,086 (51.2) | 32,142 (50.4) | 32,301 (50.1) | 32,677 (49.9) |
| Male | 24,385 (49.7) | 31,518 (48.8) | 31,664 (49.6) | 32,164 (49.9) | 32,823 (50.1) |
| Missing | 0 | 17 | 0 | 494 | 0 |
| Slept under ITN last night, n (%) | | | | | |
| Yes | 31,656 (88.3) | 48,634 (85.1) | 46,116 (96.4) | 14,626 (86.5) | 46,835 (88.8) |
| No | 4,198 (11.7) | 8,539 (14.9) | 1,711 (3.6) | 2,288 (13.5) | 5,923 (11.2) |
| Missing | 13,259 | 7,448 | 15,979[*] | 48,045[*] | 12,742[*] |
| History of fever past two weeks, n (%) | | | | | |
| Had fever | 15,709 (32.7%) | 18,575 (29.0%) | 3,724 (29.3%) | 3,459 (15.9%) | 6,892 (11.3%) |
| No fever | 32,375 (67.3%) | 45,481 (71.0%) | 9,007 (70.7%) | 18,331 (84.1%) | 54,321 (88.7%) |
| Missing | 1,029 | 565 | 51,075[*] | 43,169[*] | 4,287 |
| Geographic Zones, n (%) | | | | | |
| Lake | 12,243 (24.9) | 15,311 (23.7) | 14,962 (23.4) | 15,795 (24.3) | 15,929 (24.3) |
| Southern | 4,004 (8.2) | 2,788 (4.3) | 2,784 (4.4) | 3,674 (5.7) | 3,732 (5.7) |
| Western | 4,886 (9.9) | 6,720 (10.4) | 6,504 (10.2) | 6,119 (9.4) | 6,012 (9.2) |
| Eastern | 7,243 (14.7) | 13,272 (20.5) | 12,064 (18.9) | 10,637 (16.4) | 10,504 (16.0) |
| Southwest Highlands | 4,948 (10.1) | 6,418 (9.9) | 6,450 (10.1) | 6,501 (10.0) | 6,500 (9.9) |
| Southern Highlands | 4,137 (8.4) | 4,676 (7.2) | 4,472 (7.0) | 5,644 (8.7) | 5,640 (8.6) |
| Northern | 6,227 (12.7) | 9,036 (14.0) | 9,010 (14.1) | 8,831 (13.6) | 9,407 (14.4) |
| Central | 5,425 (11.0) | 6,400 (9.9) | 7,560 (11.8) | 7,758 (11.9) | 7,776 (11.9) |
| School elevation, n (%) | | | | | |
| 0–<750m asl | 13,990 (28.5) | 18,819 (29.1) | 17,772 (27.9) | 18,006 (27.7) | 18,123 (27.7) |
| 751–<1250m asl | 18,150 (37.0) | 23,027 (35.6) | 23,102 (36.2) | 24,623 (37.9) | 24,834 (37.9) |
| 1251–1750m asl | 14,520 (29.6) | 19,356 (30.0) | 19,508 (30.6) | 19,010 (29.3) | 18,917 (28.9) |
| >1,750m asl | 2,453 (5.0) | 3,419 (5.3) | 3,424 (5.4) | 3,320 (5.1) | 3,626 (5.5) |

Key: Median (IQR= Interquartile range); n (%);

[*]Missing values (denoted by asterisks) arise from the study design, where only subsamples of the surveyed school-aged children (SAC) were interviewed.

During 2015–2023, malaria prevalence declined both in children who reported using and not using an ITN, the night before the survey (Table 1). However, in each survey, prevalence was significantly lower among children who reported using an ITN.

School children with a history of fever in the two weeks before the survey had higher malaria prevalence compared to those who reported having no history of fever in all the five surveys. Malaria prevalence among school children declined progressively from 2015 to 2021 across all geographical zones. The Lake zone consistently reported the highest malaria prevalence, ranging from 38.7% (95%CI:37.8%–39.5%) in 2015 to 22.3% (95%CI:21.6%–22.9%)

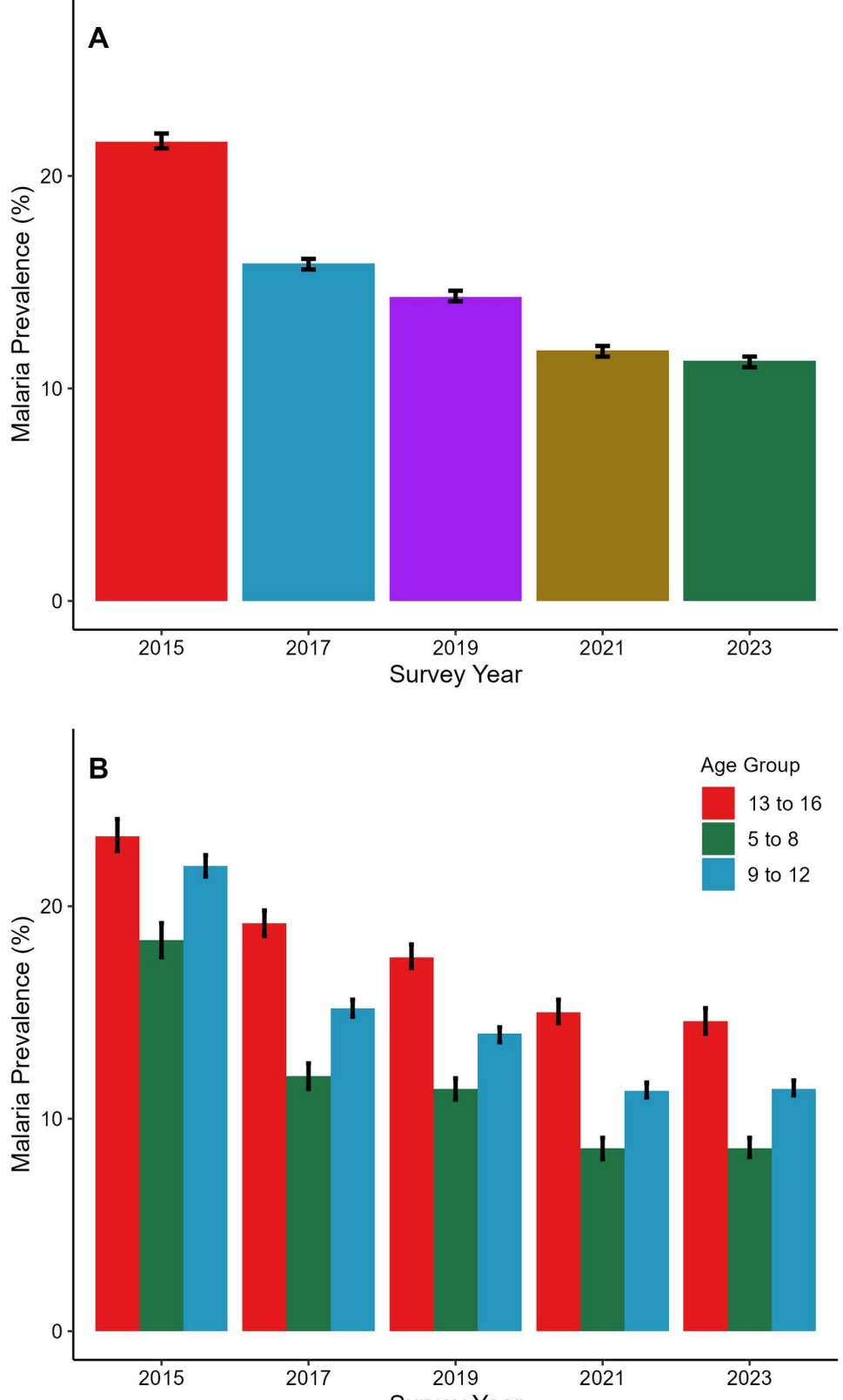

**Fig 2. Malaria infection prevalence across different age groups and survey years is presented.** Fig 2A shows trends in malaria prevalence over the years (2015–2023) among all surveyed SAC in the country, while Fig 2B illustrates

age-disaggregated malaria prevalence among SAC during the same period. Error bars represent 95% confidence intervals, unadjusted for school clustering.

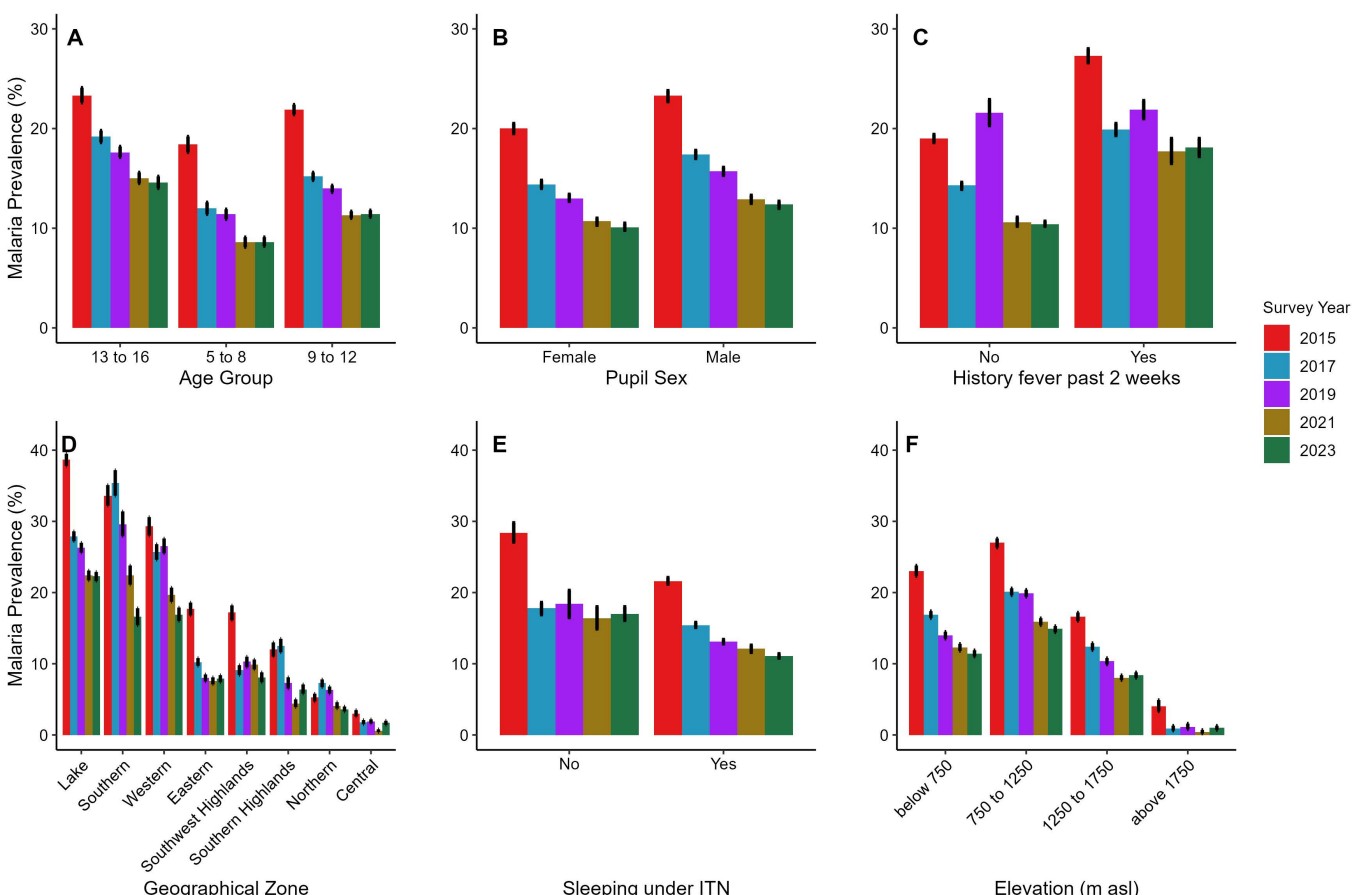

**Fig 3. Malaria prevalence over the years is presented by age (Fig 3A), sex (Fig 3B), history of fever (Fig 3C), geographical zones (Fig 3D), ITN use (Fig 3E), and elevation (Fig 3F).** History of fever and ITN use were self-reported by SAC during the interviews. Error bars indicate 95% confidence intervals, unadjusted for school clustering.

in 2023 and the Central zone consistently reported the lowest prevalence ranging from 3.0% (95%CI:2.6%–3.5%) in 2015 to 0.6% (95%CI:0.4%–0.8%) in 2021 and slightly increased to 1.7% (95%CI 1.4%–20%) in 2023 (Fig 3 and: S2 Table).

Schoolchildren living in areas between 750 and <1250 meters above sea level (asl) consistently exhibited the highest malaria prevalence, ranging from 27.0% (95%CI:26.3%–27.6%) in 2015 to 14.9% (95%CI 14.4%–15.3%) in 2023. Conversely, children residing in areas above 1750 m asl consistently reported the lowest prevalence, declining from 4.0% (95% CI 3.3%–4.9%) in 2015 to 0.4% (95% CI 0.2%–0.7%) in 2021 with a slight increase to 1.0 (0.7%–1.3%) in 2023 (Fig 3 and S2 Table).

## Multivariate analysis

The odds of malaria prevalence were higher among children aged 9–12 years old (AOR 1.20; 95% CI: 1.15–1.25, p < 0.001) and 13–16 years (AOR 1.21; 95% CI: 1.16–1.27, p < 0.001) compared to children aged 5–8 years (Table 2). Males, children had higher odds of malaria

prevalence (AOR 1.33, 95% CI: 1.29–1.38; p < 0.001) compared to females. Sleeping under ITN had lower odds of malaria prevalence (AOR 0.72; 95% CI: 0.69–0.76, p < 0.001) compared to those not sleeping under an ITN. Children with a history of fever in the past two weeks had a higher odds of malaria prevalence (AOR 1.18; 95% 1.14–1.22, p < 0.001) compared to those without a history of fever.

Table 2 indicate the odds of malaria prevalence were higher among school children in the Lake (AOR 18.75; 95% CI; 12.91–27.23, p<0.001), Western (AOR 15.52; 95% CI 10.48–22.99, p<0.001), Southern (AOR 13.09; 95% CI: 7.90–21.68, p<0.001), Southwest High lands (AOR 9.46; 95% 6.12–14.62, p<0.001) and Southern Highlands (AOR 3.08; 95%CI: 1.86–5.13, p<0.001) zones compared to Central zone. There was no statistically significant difference in

**Table 2. Crude and adjusted odds ratios of malaria prevalence in school children using multilevel mixed-effect logistic regression model in mainland Tanzania, 2015–2023.**

| Variables | COR (95% CI) | P value | AOR (95% CI) | P value |
|---|---|---|---|---|
| Age group (years) | | | | |
| 5–8 | 1 | 1 | 1 | 1 |
| 9–12 | 1.2 (1.16–1.24) | <0.001 | 1.18 (1.14–1.22) | <0.001 |
| 13–16 | 1.26 (1.22–1.31) | <0.001 | 1.22 (1.18–1.26) | <0.001 |
| Sex | | | | |
| Female | 1 | 1 | 1 | 1 |
| Male | 1.34 (1.31–1.38) | <0.001 | 1.34 (1.31–1.37) | <0.001 |
| Geographic zones | | | | |
| Central | 1 | 1 | 1 | 1 |
| Eastern | 1.5 (1.50–1.50) | <0.001 | 0.95 (0.61–1.47) | 0.81 |
| Lake | 11.33 (11.33–11.33) | <0.001 | 10.16 (7.09–14.56) | <0.001 |
| Northern | 0.59 (0.59–0.59) | <0.001 | 0.59 (0.38–0.90) | 0.015 |
| Southern | 11.46 (11.46–11.46) | <0.001 | 7.44 (4.69–11.79) | <0.001 |
| Southern Highlands | 1.13 (1.13–1.13) | <0.001 | 1.57 (0.95–2.60) | 0.081 |
| Southwest Highlands | 4.88 (4.88–4.88) | <0.001 | 8.64 (5.84–12.78) | <0.001 |
| Western | 9.93 (9.93–9.93) | <0.001 | 8.39 (5.81–12.14) | <0.001 |
| School elevation | | | | |
| 0–<750m asl | 1 | 1 | 1 | 1 |
| 751–<1250m asl | 2.99 (2.54–3.53) | <0.001 | 0.82 (0.62–1.09) | 0.175 |
| 1251–1750m asl | 1.12 (0.90–1.39) | 0.31 | 0.42 (0.31–0.57) | <0.001 |
| >1,750m asl | 0.06 (0.03–0.11) | <0.001 | 0.04 (0.02–0.07) | <0.001 |
| History of fever past two weeks | | | | |
| No | 1 | 1 | | |
| Yes | 1.2 (1.17–1.24) | <0.001 | | |
| Slept in ITN last night | | | | |
| No | 1 | 1 | | |
| Yes | 0.68 (0.65–0.71) | <0.001 | | |
| Survey Year | | | | |
| 2015 | 1 | 1 | 1 | 1 |
| 2017 | 0.80 (0.77–0.83) | <0.001 | 0.80 (0.77–0.83) | <0.001 |
| 2019 | 0.65 (0.62–0.68) | <0.001 | 0.65 (0.63–0.68) | <0.001 |
| 2021 | 0.44 (0.42–0.46) | <0.001 | 0.44 (0.42–0.46) | <0.001 |
| 2023 | 0.44 (0.34–0.57) | <0.001 | 0.44 (0.35–0.54) | <0.001 |

the odds of prevalence among school children in Eastern and Northern zones compared to the Central zone.

The odds of malaria prevalence significantly decreased among schoolchildren in 2023 (AOR 0.47; 95% CI: 0.38–0.58, p<0.001) compared to 2015. However, no statistically significant difference in the odds of prevalence was observed in 2021 (AOR 0.46; 95%CI: 0.43–0.49, p<0.001) and 2023 (AOR 0.47; 95%CI: 0.38–0.58, p<0.001) (Table 2).

## Discussion

In our study, we found malaria infection prevalence significantly declined among school children attending public primary school in mainland Tanzania during 2015–2023 but without major changes in reduction in malaria infection prevalence in 2021 and 2023. Older school children (9–16) had a higher malaria infection prevalence compared to younger children (5–8 years) and males had a higher prevalence compared to females. Children sleeping under ITNs had a lower malaria prevalence compared to those who did not, and a history of fever was associated with higher malaria prevalence. There was a significant difference in heterogeneity in malaria infection prevalence in school children living in different geographic zones. Children in the Lake, Western and Southern zones had a higher prevalence compared to other zones. The observed declining malaria prevalence among school children provides valuable insight of the effort undertaken to control malaria in mainland Tanzania and underscores the importance of a comprehensive malaria control intervention package, continued surveillance and stratified interventions to maximize impact and achieve malaria elimination.

Malaria prevalence among school children reported in our study is higher than the prevalence among children under five years (8.1% in 2022) [20] and pregnant women (6.1% in 2023) in mainland Tanzania [23]. Malaria prevalence was notably higher among older children (aged 13–16 years) compared to younger children, which is consistent with findings from other studies [24–26]. This could be attributed to behavioral factors associated with older children, such as increased outdoor activities during peak mosquito biting hours, which might not be as effectively mitigated by ITN usage alone. Despite the observed decline, the higher malaria prevalence among male children, consistently throughout the study period, suggests potential gender-specific behavioral or biological factors that warrant further investigation to tailor more effective interventions to address outdoor and residual malaria transmission and advocacy to address potential gender-specific behavioral factors. This raises important questions about the need for malaria control policies specifically tailored to school children.

To increase access to ITNs in the community, Tanzania introduced universal coverage campaign (UCC) distribution in 2010 followed by mass replacement campaign (MRC) in 2015 and continuous ITN distribution through School Net Programme (SNP) in 2012 and Reproductive Child Health (RCH) 2015. Since the inception of SNP, it has become one of the major distribution channels for ITN across the country and has demonstrated a significant contribution in the reported ITN ownership and usage findings in our study. Malaria prevalence was significantly lower among children who reported using an ITN compared to those who did not, which was consistent across all survey years. This underscores the critical role of continued distribution and promotion of ITN use as a cornerstone of malaria prevention strategies. However, despite the protective effects of ITNs on malaria prevalence, our study found that the malaria prevalence among school children who slept under an ITN was not negligible. A qualitative study in Dar es Salaam City in Tanzania identified several factors contributing to high malaria prevalence, including late bedtimes, improper use of ITNs, shared sleeping arrangements, small-sized beds, and damaged nets with holes [27]. These findings highlight ongoing issues with the physical and chemical integrity of ITNs, usage patterns, and potential resistance to the insecticides used in these nets. Despite the overall decrease in malaria

prevalence and the clear protective effect of ITN usage, these challenges underscore the need for enhanced strategies in ITN distribution, education on proper usage, and the development and introduction of next-generation ITNs against resistant mosquito strains. Such efforts are crucial to maximize the protective benefits of ITNs and further reduce malaria prevalence in Tanzania.

The persistent geographic heterogeneity in malaria prevalence, with significantly higher prevalence in the Lake, Western, and Southern zones, highlight the role of ecological and environmental factors in malaria transmission. These areas likely provide more favorable breeding conditions for mosquitoes, coupled with socio-economic and cultural factors that might affect exposure risks. It is also notable that the Lake, Western, and Southern zones with highest malaria prevalence also share border with several countries and have different environmental and cultural background, highlights the need for expansion of cross border malaria control policies and coordination. To address this geographic heterogeneity, mainland Tanzania introduced malaria stratification to inform targeted interventions at sub-national level using data generated from SMPS and health facility data [15,28].

The significant decrease in malaria prevalence among school children over the years is encouraging; however, the lack of substantial decline between 2021 and 2023 mirrors the prevalence observed in children under five years (7.5% in 2017 to 8.1% in 2022) [20] which signals plateau of malaria prevalence in these two age groups in recent years. This plateau may indicate that current mix of control interventions have reached their effectiveness limits in some contexts, or that new challenges, such as mosquito resistance that have been observed in the country [29] and/or changes in mosquito composition and behavior [30], could explain the situation. The insights from our study stress the importance of continuous surveillance and adaptable malaria control strategies that consider local ecological, behavioral, and socio-economic conditions. Further, our findings advocate for targeted interventions tailored to specific age groups, sex, and geographic zones to enhance the effectiveness of malaria control efforts in Tanzania.

The scope of SMPS was limited to public schools, potentially overlooking the situation in private schools. Although, the SMPS could not establish whether the same children were sampled across survey years or tracked repeated cases of malaria infection which may lead to intra-child correlation, it is expected to have minimal impact on the analysis due to random sampling. We relied on self-reported ITN use and history of fever, which could be biased. Absences to school due to illness or other reasons on survey days might have led to underestimating the true prevalence. Malaria RDTs might have overestimated prevalence in regions with high infection rates, as these tests cannot differentiate between recent and older infections. An evaluation of the diagnostic performance of mRDTs (HRP2 and pan-parasite lactate dehydrogenase) used in the 2017 surveys against polymerase chain reaction (PCR) for detecting *P. falciparum* showed moderate sensitivity and high specificity for screening asymptomatic schoolchildren in higher transmission zones [31]. However, the effectiveness of mRDTs decreased in low transmission settings, likely due to the presence of low parasitemia infections. The overall prevalence of *pfhrp2* and *pfhrp3* gene deletions was found to be low, infections by strains with these deletions were predominantly found in the Kagera region, a high transmission area [32]. These findings suggest that the choice of mRDT for future surveys should be informed by these variations in diagnostic performance.

Our study shows a significant decline in malaria prevalence among Tanzanian school children from 2015 to 2023, highlighting the impact of robust malaria control initiatives like the widespread ITN use made available in part through the SNP and targeted interventions according to malaria transmission risks. Despite this success, the observed plateau in

prevalence reduction between 2021 and 2023 suggests maintaining the implementation of existing targeted interventions and consideration of scale-up of recommended initiatives such as provision of intermittent preventive treatment in school-aged children (IPTsc), and continual monitoring of insecticide resistance, behavioral factors that compromise ITN efficacy, and outdoor mosquito bites which contribute to malaria transmission. Geographic and demographic disparities in malaria prevalence underscore the need for strategies that address the specific conditions of high malaria prevalence zones and populations, including cross-border areas with heightened transmission risks. The persistent higher prevalence among older and male children indicates behavioral risks that require focused prevention strategies. Ultimately, sustaining the gains achieved and advancing toward malaria elimination in Tanzania will require ongoing surveillance, adaptable interventions, and a commitment to addressing both the biological and socio-economic determinants of malaria transmission.

## Supporting information

**S1 Fig.** Multistage sampling procedures for Wards, Schools and School Children. The selection of wards, schools, and school children followed a multistage, stratified, proportional probability-to-size sampling procedure. Fig. **A** illustrates the selection of wards and schools, Fig. **B** depicts the selection of female children, and Fig. **C** shows the selection of male children. These figures were created using PowerPoint, and the icons representing male and female figures are licensed under the Creative Commons CC0 1.0 Universal Public Domain Dedication. Links to these icons are as follows: File: Woman - The Noun Project.svg - Wikimedia Commons, File: Man - The Noun Project.svg - Wikimedia Commons.
(TIF)

**S1 Table.** Tanzania SMPS data collection period.
(DOCX)

**S2 Table.** Trend malaria prevalence with different background characteristics, 2015-2023.
(DOCX)

**S1 Text.** Checklist of Inclusivity in global research questionnaires.
(DOCX)

## Acknowledgments

The SMPS study was conducted by the Tanzania Ministry of Health through the National Malaria Control Program, in collaboration with research Institutions and academia namely; National Institute for Medical Research, Ifakara Health Institute, Muhimbili University of Health and Allied Sciences, University of Dar es Salaam, Sokoine University of Agriculture, Tanzania Food and Nutrition Center. The authors are grateful to all Ministry of Health, President's Office Regional Administration and Local Government, Ministry of Education, Science and Technology, institutions, partners, field team, data entry clerks, and investigation team and individuals who contributed to this school survey including surveyed school children for their voluntary participation in the survey. Authors would like to extend their much appreciation to Julieth Silao, David Dadi, Victor Alegana, Samson Kiware, Benjamin Kamala, Brigita Msofe, Witness Saitot, Abdallah Lusasi, Anna David, Pendael Machafuko, Felista Mwingira, Bwire Wilson, Charles Dismas Mwalimu, Agnes Mpinga, Fidelis Mgohamwende, Humphrey Mkali and Wiggins Aaron for their valuable inputs and advise.

## Author contributions

**Conceptualization:** Frank Chacky, Susan F Rumisha, Patrick GT Walker, Prosper Chaki, Fabrizio Molteni, Billy Ngasala, Bruno P. Mmbando, Robert W. Snow.

**Data curation:** Frank Chacky, Joseph T. Hicks, Mbaraka John Remiji, Susan F Rumisha, Bruno P. Mmbando.

**Formal analysis:** Frank Chacky, Joseph T. Hicks, Mbaraka John Remiji, Susan F Rumisha, Patrick GT Walker, Bruno P. Mmbando.

**Funding acquisition:** Frank Chacky, Sijenunu Aaron, Samwel L. Nhiga, Erik Reaves, Naomi Serbantez, Fabrizio Molteni, Achyut KC, Robert W. Snow.

**Investigation:** Frank Chacky, Susan F Rumisha, Prosper Chaki, Sijenunu Aaron, Billy Ngasala, Bruno P. Mmbando.

**Methodology:** Frank Chacky, Mbaraka John Remiji, Susan F Rumisha, Patrick GT Walker, Fabrizio Molteni, Billy Ngasala, Achyut KC, Bruno P. Mmbando, Robert W. Snow, Jean-Pierre Van Geertruyden.

**Project administration:** Frank Chacky, Susan F Rumisha, Prosper Chaki, Sijenunu Aaron, Samwel L. Nhiga, Naomi Serbantez, Fabrizio Molteni, Billy Ngasala.

**Resources:** Frank Chacky, Prosper Chaki, Sijenunu Aaron, Naomi Serbantez, Fabrizio Molteni, Billy Ngasala, Robert W. Snow, Jean-Pierre Van Geertruyden.

**Software:** Joseph T. Hicks, Mbaraka John Remiji, Bruno P. Mmbando.

**Supervision:** Frank Chacky, Susan F Rumisha, Prosper Chaki, Sijenunu Aaron, Samwel L. Nhiga, Fabrizio Molteni, Billy Ngasala, Achyut KC, Robert W. Snow, Jean-Pierre Van Geertruyden.

**Validation:** Susan F Rumisha, Patrick GT Walker, Achyut KC, Bruno P. Mmbando, Robert W. Snow, Jean-Pierre Van Geertruyden.

**Visualization:** Frank Chacky, Joseph T. Hicks, Susan F Rumisha, Patrick GT Walker, Erik Reaves, Achyut KC, Bruno P. Mmbando, Robert W. Snow, Jean-Pierre Van Geertruyden.

**Writing – original draft:** Frank Chacky.

**Writing – review & editing:** Frank Chacky, Joseph T. Hicks, Mbaraka John Remiji, Susan F Rumisha, Patrick GT Walker, Prosper Chaki, Sijenunu Aaron, Samwel L. Nhiga, Erik Reaves, Naomi Serbantez, Fabrizio Molteni, Billy Ngasala, Achyut KC, Bruno P. Mmbando, Robert W. Snow, Jean-Pierre Van Geertruyden.

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
